# Electrostatic Potential Topology for Probing Molecular Structure, Bonding and Reactivity

**DOI:** 10.3390/molecules26113289

**Published:** 2021-05-29

**Authors:** Shridhar R. Gadre, Cherumuttathu H. Suresh, Neetha Mohan

**Affiliations:** 1Department of Chemistry, Interdisciplinary School of Scientific Computing, Savitribai Phule University, Pune 411007, India; 2Chemical Sciences and Technology Division, CSIR-National Institute for Interdisciplinary Science and Technology, Trivandrum 695019, India; neethamohan@gmail.com

**Keywords:** molecular electron density (MED), molecular electrostatic potential (MESP), scalar fields, topology, critical points, substituent constants, reaction mechanism, electrophilic attack, cycloaddition, aromaticity, lone pair, non-covalent interaction

## Abstract

Following the pioneering investigations of Bader on the topology of molecular electron density, the topology analysis of its sister field viz. molecular electrostatic potential (MESP) was taken up by the authors’ groups. Through these studies, MESP topology emerged as a powerful tool for exploring molecular bonding and reactivity patterns. The MESP topology features are mapped in terms of its critical points (CPs), such as bond critical points (BCPs), while the minima identify electron-rich locations, such as lone pairs and π-bonds. The gradient paths of MESP vividly bring out the atoms-in-molecule picture of neutral molecules and anions. The MESP-based characterization of a molecule in terms of electron-rich and -deficient regions provides a robust prediction about its interaction with other molecules. This leads to a clear picture of molecular aggregation, hydrogen bonding, lone pair–π interactions, π-conjugation, aromaticity and reaction mechanisms. This review summarizes the contributions of the authors’ groups over the last three decades and those of the other active groups towards understanding chemical bonding, molecular recognition, and reactivity through topology analysis of MESP.

## 1. Introduction

The computation of accurate molecular wave functions and their analysis has remained a challenge for theoretical chemists over the years due to the enormous dimensionality of the problem. In particular, the non-relativistic wave function is dependent on 3N spatial coordinates, N being the number of nuclei in the molecule. The computation of accurate molecular wave functions has become practically feasible in the last few decades owing to the advent of powerful computers and algorithmic advances. Attempts to obtain a condensed description of the large details buried in molecular wave function has led to the use of three-dimensional (3D) molecular scalar fields. Several 3D molecular scalar fields derived from the 3N-dimensional wave function have been explored by theoretical as well as experimental chemists to understand the structure, bonding, and reactivity patterns of molecules. Many of these 3D scalar fields, such as the molecular electron density (MED) in position and momentum spaces and the molecular electrostatic potential (MESP), are experimentally amenable, thereby providing a vital bridge between experiment and theory. The essential features of the scalar fields are routinely captured by mapping their topology, which involves identifying and characterizing critical points (CPs), generating isosurfaces and gradient paths of the scalar field under scrutiny [1,2,3]. The topographical features of any scalar function can be described in terms of its first and second-order partial derivatives or in terms of the number and nature of critical points (CPs). The CP of a three-dimensional (3D) scalar field described by a function f(x_1_, x_2_, x_3_) is defined as the point P at which all three first-order partial derivatives vanish, i.e., for i = 1, 2, 3.
(1)$${\nabla_{i}f\left(x_{1},x_{2},\ldots ,x_{n}\right)|}_{P}={\frac{\partial f}{\partial x_{i}}|}_{P}=0$$

The CP is further characterized [1,2,3] by the evaluation of the eigenvalues of the corresponding Hessian matrix H_ij_ at the point P, where
(2)$$H_{\mathrm{ij}}={\left[\frac{\partial^{2}f}{\partial x_{i}\partial x_{j}}\right]}_{P}$$

A CP is designated as an ordered pair (r, ω). Here, r designates the rank, i.e., the number of non-zero eigenvalues of the function at the CP, and ω is the signature or the algebraic sum of the signs of eigenvalues of the Hessian matrix [1]. A CP for which at least one of the eigenvalues of the Hessian matrix is zero is termed a degenerate CP. Thus, for a 3D scalar field, there can be four types of non-degenerate CPs [1,3], viz. (3, +3), (3, −3), (3, +1), and (3, −1). Here, a (3, −3)-type CP corresponds to a local maximum where all of the eigenvalues of the Hessian matrix become negative. The saddle points are denoted as (3, +1) and (3, −1). The (3, +3)-type is a local minimum [1,3]. The Laplacian of the function f can be expressed as the sum of the eigenvalues of the Hessian matrix, viz.
(3)$$\nabla^{2}f=\lambda_{1}+\lambda_{2}+\lambda_{3}$$
where the λ_i_s indicate the eigenvalues of the Hessian matrix.

The 3D scalar field of molecular electron density (MED) assumes a central role in density functional theory (DFT) [4], which employs the 3D electron density, ρ(**r**), as the basic variable replacing the multidimensional wave function Ψ(**x**_1_, **x**_2_…**x**_N_). The MED [2], ρ(**r**), describes the probability distribution of electrons in the system and can be extracted from the corresponding wave function by a suitable integration followed by the substitution **r** = **r**_1_, viz.
(4)$$\rho \left(r\right)=\;N\underset{\sigma }{\sum }\int {|\Psi \left(x_{1,}\;x_{2,\;}\ldots x_{N}\right)|}^{2}d^{3}r_{2}\ldots {d^{3}r_{N}|}_{x1=x}$$
where **x**_i_ denotes the combined spatial (**r**) and spin (σ) coordinate of the electron i, and N is the number of electrons. Powerful methods that exploit the scalar field of MED for probing fundamental chemical concepts such as the chemical bond, molecular structure, π-delocalization, lone pairs, etc., were established by Bader and co-workers [5,6,7,8,9]. The treatise, *Atoms in Molecules: A Quantum Theory* [1], offers a protocol for partitioning molecular properties into atomic contributions based on the topological properties of MED. Bader’s quantum theory of atoms in molecules (abbreviated as QTAIM or AIM) starts by locating and characterizing the MED CPs. In the AIM parlance, every pair of bonded atoms exhibits characteristic (3, −1) saddle point between them, known as a bond critical point (BCP). The (3, +1) saddle point, characteristic of a ring structure, is termed as the ring critical point (RCP). The (3, +3) saddle point, termed a cage critical point (CCP), signifies a cage structure. The RCP is a one-dimensional attractor while CCP is a three-dimensional repeller [1]. The path generated by the gradient vector of ρ(**r**) starting from an initial point is termed as a gradient path. The gradient paths always originate at a CP and terminate at another CP [1]. A unique gradient path of maximum electron density traversing through BCP connecting the two interacting atoms is referred to as the Atomic Interaction Line (AIL) or Bond Path [1,10]. A network of bond paths linking the nuclei of bonded atoms of a molecule at its equilibrium geometry with the associated critical points is known as the molecular graph. The Laplacian of the electron density designated as ∇^2^ρ(**r**) was another scalar field investigated by Bader and co-workers [1]. This is an integral part of AIM theory which measures to what extent ρ(**r**) is locally concentrated/depleted. If ∇^2^ρ(**r**) < 0, the MED is locally concentrated, indicating a sharing of electron density by two atoms, viz. a covalent bond, whereas ∇^2^ρ(**r**) > 0 indicates that the MED is locally depleted.

Cremer and Kraka [11,12] pointed out that it is not possible to detect all chemical bonds of a molecule if bonding is solely described by the degree of charge accumulation in the internuclear region. Alternatively, they suggested the use of total energy density, *h*(**r**), which is the sum of kinetic energy density, *g*(**r**), and the potential energy density, v(**r**), as a useful parameter to distinguish covalent bonding and closed-shell interactions such as electrostatic, ionic or dispersion interactions as well as other non-covalent interactions. Based on these observations, Cremer and Kraka proposed the necessary and sufficient conditions for covalent bonding, which are as follows: (A) The existence of a bond path between the bonded atoms and (B) the local energy density at the BCP is stabilizing, i.e., *h*(**r**) < 0. AIM parameters were also used to define the bond strain, bond ellipticity, bond order, homoaromaticity, etc. [1]. The basic assumption of QTAIM that the local maxima of electron density occur only at the nuclear positions is not universal [13,14]. Gatti and co-workers [15,16] showed that some molecules indeed exhibit non-nuclear maxima, i.e., local maxima of electron density at points other than the nucleus. Non-nuclear maxima have been noted especially in metals, semiconductors and at the positions of defects in crystals. The universal bond path criterion to determine whether two atoms are bonded has been a matter of detailed scrutiny [17,18,19,20,21]. However, Bader [6,22] underlined that bond paths, being a measurable property of the system, should be used as a topological proof of the bonding of atoms but not be taken as the presence of a chemical bond. The latter “is neither measurable nor susceptible to theoretical definition and means different things to different people” [22].

The sister scalar field of MED viz. the molecular electrostatic potential (MESP) has been used in the chemical literature for a long time as a tool for exploring the structure and reactivity in molecules [23,24,25,26]. The MESP, V(**r**), at a given point **r** is defined as the work carried out in bringing a test positive charge from infinity to the reference point **r** and has a connection with the energy. The MESP is given by the equation:(5)$$V\left(r\right)=\overset{\mathrm{Na}}{\underset{A}{\sum }}\frac{Z_{A}}{\left|r-R_{A}\right|}-\int \frac{\rho \left(r^{\prime }\right)d^{3}r^{\prime }}{\left|r-r^{\prime }\right|}$$
where Z_A_ is the charge of the nucleus A located at **R**_A_, Na being the number of nuclei and **r^’^**, the dummy integration variable. The first term on the r.h.s. (right hand side) of Equation (5) is the bare nuclear potential (BNP), while the second term represents the electronic contribution. The MESP can attain negative, zero, or positive values, in contrast to the electron density, which takes only non-negative values. In the regions of high electron concentration, the second term dominates over the first term and MESP exhibits negatively valued minima and saddles, whereas the regions with positive MESP reflect the dominance of the nuclear contribution. It was shown by Pathak and Gadre [27] that MESP does not exhibit non-nuclear maxima unlike the MED. Popelier and Brémond observed geometrically faithful homeomorphisms between the topology of ρ(**r**) and BNP [28].

Early works of Scrocco, Tomasi and the Pullmans [29,30,31,32] established MESP as a guiding scalar field for understanding the reactive behavior of molecules. Later, Politzer and co-workers popularized the qualitative use of MESP to predict the molecular sites amenable to electrophilic and nucleophilic attack [23,33,34,35]. They employed the positive and negative regions generated by texturing the MESP on a molecular surface for identifying the sites of nucleophilic and electrophilic attack, respectively [23,36,37,38]. These pioneering studies led to extensive application of MESP to describe chemical properties such as bonding, inductive and resonance effects, and prediction of the protonation sites of organic molecules including nucleic acid bases as well as hydration sites of large molecules [29,33,34,39,40]. MESP has also been useful for exploring molecular recognition in biological systems. Náray Szabó and co-workers [41,42] tried to understand molecular recognition qualitatively in terms of MESP, where the enzyme active site and ligand are represented as an electrostatic lock and key, respectively. Gadre and Bendale [43] explored the qualitative similarity between contour maps of MESP and bare nuclear potential. Despite these qualitative uses for probing reactivity, MESP remained quite away from rigorous topological analysis.

Gadre and co-workers made early pioneering contributions towards understanding the topological features of MESP. They also supplemented these efforts by developing algorithms and software for the computation and topology mapping of MESP. The forthcoming sections will summarize their studies related to the topology description of molecules in the scalar field MESP and also those of the other active groups. The quantum chemical topology description of molecules in the scalar field of electron density is well documented in the literature [5,6,7,22,44,45] and the QTAIM perspective of chemical bonding is elaborately described by Popelier [46]. The topological analyses of electron localization functions (ELF) [47,48] that magnetically induced molecular current distributions [49] and Ehrenfest force field [50,51] have also provided information on chemical bonding and molecular structure in great detail. Since the focus of the present review is on MESP, we do not attempt to review such studies.

## 2. MESP Topology: Early Basic Studies and Algorithm Development

In this section, we discuss the earlier basic studies conducted by Gadre’s research group towards understanding MESP topography along with their efforts to develop algorithms for locating MESP CPs. Weinstein et al. [52] proved the nonexistence of maxima in the electrostatic potential of spherically symmetric atomic systems in 1975. For this purpose, they employed the Poisson equation, in spherical polar coordinates, viz.
(6)$$\frac{d^{2}V\left(r\right)}{{\mathrm{dr}}^{2}}+\frac{2}{r}\frac{\mathrm{dV}\left(r\right)}{\mathrm{dr}}=4\pi \rho \left(r\right)$$

For a minimum to occur in V(r), its first order derivative must vanish and its second order derivative must be negative. Since ρ(r) on the r.h.s. cannot be negative, the occurrence of local maximum for atomic ESP is ruled out. Sen and Politzer [53] extended this work to the case of a monoatomic anion. The electrostatic potential V(r) for this case is given by
(7)$$V\left(r\right)=\frac{Z}{r}-\int \frac{\rho \left(r^{\prime }\right)}{\left|{r\;-\;r}^{\prime }\right|}d^{3}r\prime$$

The anionic V(r) necessarily has a negative minimum at a finite radial distance, r_m_. The potential at this point, V(r_m_), is entirely due to the net ionic charge and the distance, r_m_, can be interpreted as the corresponding anionic radius. They found a good correlation between the radii of monoatomic anions of main group elements estimated using self-interaction-corrected local spin density (SIC-LSD) approximation and the corresponding literature values. This approach was proved to be valid for transition metals systems as well [54]. Sen and Politzer further proposed that the magnitude of the electrostatic potential, V(r_m_), directly reflects the strength of anionic interaction with a cation [53,54]. They demonstrated this conjecture using alkali metal salts of halides F^−^, Cl^−^, Br^−^ and I^−^, yielding good correlation between lattice energies and the corresponding V(r_m_) values.

The generalization of “no maximum” theorem to molecules took almost 15 years. Gadre and Pathak proved in 1990 that the property of nonexistence of local maxima is indeed general and holds true for molecular systems [55] at all points in space, except at the nuclear sites. Their proof is based on the classical Poisson equation in three dimensions, viz.
(8)$$\nabla^{2}V\left(r\right)=4\pi \rho \left(r\right)$$

Since the r.h.s. of Equation (8) is always positive and ∇^2^V(**r**) ≤ 0 for a maximum to occur at **r**, it follows that MESP cannot exhibit non-nuclear maxima for the ground as well as excited states of molecules. A discussion regarding the case of degenerate CPs can be seen in Refs. [56,57]. Gadre and Pathak [27] further showed that the MESP of negative ions exhibits at least one directional negative-valued minimum along every ray starting out from a nuclear position. A surface, S, encompassing all the nuclei would exist for such negative molecular ions, satisfying the equation,
(9)∇V(r).dS=0

Employing Gauss’ theorem, the integral ∇V(r).dS over S can be converted to a volume integral over the enclosing volume τ leading to
(10)$$\underset{\Omega }{\int }\nabla V\left(r\right).dS=\underset{\tau }{\int }\nabla^{2}V\left(r\right)d^{3}r\;=0$$

This implies that the net charge of the anion resides outside the minimal surface, S. Gadre and co-workers used the rich topology of MESP for visualizing anionic sizes, shapes, nucleophilicity, and softness [58,59]. Gadre, Kölmel and Shrivastava [60] showed that MESP maps of anions provide a practical tool to predict various chemical properties including anionic shapes and sizes, nucleophilicity, softness and anisotropies in electrostatic interactions of anions with ions/molecules. They also found that for every molecular anion, there exists a girdle of minima and saddle points surrounding the nuclear skeleton. These early works stimulated rigorous efforts for bringing out the detailed general topological features of MESP [58,61].

Gadre and co-workers defined and characterized the CPs of MESP from a detailed study of the MESP maps of diatomic and small polyatomic molecules. The salient features of MESP topography emerging from these studies may be summarized as follows. 

The topology of MESP is rich with CPs of (3, +3), (3, +1) and (3, −1) type. Akin to MED topology, the MESP topography also exhibits positive (3, −1) CP between a pair of bonded atoms (Figure 1 and Figure 2). However, as seen earlier, MESP does not show non-nuclear maxima [55,57]. This is in contrast to MED, wherein non-nuclear maxima are seen [15,16] in several molecular systems.

MESP can attain negative values in contrast to MED, which is strictly non-negative. Lone pairs and π bonds are manifested (Figure 2) as negative-valued local minima in MESP viz. (3, +3) CPs, which are generally absent in the topology of ρ(**r**).

The presence of a (3, +1) saddle point lying between the (3, +3) CPs is an additional feature displayed by the MESP topology which is also generally absent in the scalar field of MED. However, (3, +1) and (3, +3) saddle points do occur as signatures of ring and cage structures, respectively, in the scalar field of MESP (Figure 3). Thus, the (3, +1) CP is not always indicative of ring structure in MESP unlike in MED.

The occurrence of degenerate CPs is a noteworthy feature in the MESP topology. Their occurrence in the MED topology has earlier been discussed at length by Popelier [62]. Even doubly degenerate (1, −1) CPs are observed in the topology of the Laplacian of the electron density [63]. In MESP topology, non-isolated degenerate CP is generally found to occur in the linear, neutral, as well as negatively charged molecules, e.g., HF, HCl, C_2_H_2_, and OH^-^ (see Figure 4). Degenerate CPs observed in the topology of ρ(**r**) are strictly isolated ones, leading to instability in the molecular structure while non-isolated degenerate CPs are not observed in the ground state topology of ρ(**r**).

One may wonder whether the level of theory and basis set affect the MESP topological features substantially. Following the works of Luque et al. [64,65,66], Gadre and co-workers investigated the basis set dependence of MESP topology [67], employing substituted benzenes as test cases. They concluded that a large basis set incorporating polarization functions seems to be adequate for faithfully representing the MESP CPs. In a further study, Kulkarni [68] concluded that the inclusion of electron correlation does not generally affect the nature and number of MESP CPs.

With the widespread qualitative and quantitative applications of MESP, Gadre and co-workers [69,70,71] developed serial and parallel algorithms based on rigorous bounds and strategies for fast and accurate evaluation of MESP using Gaussian basis functions. The most time-consuming part in the evaluation of MESP is the computation of the electronic contribution, viz. the second term on the r.h.s. of Equation (5). For this, the analytical expressions for the individual integrals are written as the product of a point-independent and a point-dependent part, allowing the algorithm to be divided into pre-processing and processing parts. Rigorous upper bounds were applied to the analytical expression of the individual integrals at the pre-processing stage in order to filter out the numerically insignificant integrals. The ingenious application of rigorous bounds resulted in reducing the complexity of the calculation from ~N^2^ to ~N, N being the number of basis functions used. Restructuring of this algorithm using the concept of SHELLS and more rigorous bounds to enhance sequential power and load balancing strategies in parallelization allowed efficient computation of MESP of large molecules [58]. Gadre and co-workers further proposed a comprehensive topographical approach for the treatment of molecular electrostatics in terms of MESP, molecular electrostatic field (MEF) as well as the location and characterization of CPs. It is also worth mentioning that a general algorithm was developed by Malcom et al. [72] to locate critical points in the topological analysis of general 3D molecular scalar fields.

Lebeouf et al. [73] explored the Poincaré–Hopf (PH) relationship connecting the electronic structure elements (given by CPs) and electrostatic reactivity via the MESP topology. The Poincaré–Hopf (PH) relation [74] offers a link between the number of CPs and the long-range behavior of MESP and is defined in terms of the number of nondegenerate CPs as:(11)$${\;n}_{+3}-n_{+1}+n_{-1}-n_{-3}=\chi$$
where χ is the corresponding Euler characteristic. The numbers n_+3_, n_+1_ and n_−1_ denote the number of (3, +3), (3, +1) and (3, −1) CPs for MESP, respectively. For MESP, the Euler characteristic, χ can take positive, zero as well as negative values. Lebeouf et al. [56] offered an interpretation of the Euler characteristic of MESP in terms of asymptotic decay by texturing MESP on a large sphere centered at the center of mass of the molecule and the sign of MESP checked on the spherical grid. The numbers of negative and positive asymptotic decays are counted to obtain the number of asymptotic maxima and asymptotic minima designated as n_-_ and n_+_, respectively. The Poincaré–Hopf (PH) relation for MESP thus takes the form [56]:(12)$$n_{+3}-n_{+1}+n_{-1}-n_{-3}=n_{-}-n_{+}$$

The PH relation does not ensure a sufficiency check but provides only a necessary condition for ensuring that all the CPs of a molecular system are mapped. For instance, if a CP each of the types (3, +3) and (3, +1) is missed, the relationship would still hold true. Balanarayan and Gadre [75] reported an algorithm for locating the CPs of a 3D scalar field based on a ray search from the surface extrema of appropriately defined atom-centered spheres and tested it on the MESP and MED of several molecules. They also attempted a topological interpretation of the PH relation and a stronger local check for the consistency of the number CPs found. Though this local relationship did not provide a sufficiency check, it could give indications to where the CPs could be missed. Later, Roy et al. [76] proposed a general definition of the Euler characteristic and Poincaré–Hopf relation for MESP based on the gradient vectors of this scalar field on a spherical grid. Here, the Euler characteristic is derived by counting the number of inward and outward pointing regions of the MESP gradient. They utilized a shrinking sphere strategy for locating the MESP CPs of a molecule efficiently using the change in the number of inward and outward pointing regions over local and global spheres.

The problem of computation of MESP and its CPs for large molecular systems at the ab initio level is still formidable. For this purpose, Gadre et al. [77]. proposed a simple and novel methodology, named the molecular tailoring approach (MTA). In this method, the system under investigation is divided into smaller, overlapping fragments and SCF computations on them are carried out to obtain the corresponding density matrices (DM). These fragment DMs are then ‘stitched’ to synthesize the DM of the parent molecular system. This scheme was tested out for a number of systems to show that the MESP and MEF obtained by MTA closely resemble the actual ones [78,79]. MESP mapping of very large molecules/clusters is enabled by the use of MTA. Incidentally, MTA methodology was extended further for geometry optimization and calculation of IR spectra of large molecules/clusters [80,81,82]. The insights provided by MESP topology were also used for building geometries and estimating binding energies of weakly bound molecular complexes.

## 3. Recent Studies on MESP Topology and Algorithm Development 

In spite of the advances in computational hardware, the mapping MESP topography of large molecules at higher levels of theory has remained a formidable task. Yeole and Gadre [83] proposed a sequential scheme for economically mapping the complete topography of large molecules, which builds up the CPs of a molecule from the scalar field of BNP to MED and then to MESP. Yeole et al. [84] further improved this algorithm and coupled it with the deformed-atoms-in-molecules (DAM) method [85] for rapid and accurate evaluation of MED and MESP and respective gradients. MTA was invoked for the rapid topology mapping of large systems and the completeness of this mapping is checked by applying the Poincaré–Hopf relation. Computer programs for locating and characterizing the CPs of BNP, MED and MESP such as WebProp [86] and DAMQT 2.1.0 [87] (incorporating the topographical analysis of MED and MESP) were developed. Indigenously developed programs for visualization of CPs and isosurfaces include UNIVIS-2000 [88] and MeTA studio [89].

The definition of a lone pair in the literature is generally based on the molecular orbital (MO) picture. Since the MOs of a molecule are not uniquely defined, it is desirable to have a more robust definition based on MED or the quantities derived therefrom. Recently, Kumar et al. [90] employed MESP topology to provide a clear-cut definition of lone pairs. It has been known that local MESP minima ((3, +3) CPs) indicate the presence of lone pairs, π-bonds, or other pockets of electron concentration. The following two criteria were proposed to distinguish lone pair regions from other types of electron localization. (i) The magnitude of the eigenvalue at the CP that corresponds to the lone pair is numerically greater than 0.025 a.u. (atomic unit) and (ii) the eigenvector associated with the largest eigenvalue of the CP nearly points in the direction of the atom on which it is localized (angle < 5°). The (3, +3) CPs of water molecule-denoting lone pairs, for instance, have the largest eigenvalue of 0.14 a.u. and the corresponding eigenvector makes an angle of 1.4° with the line joining the MESP minimum with the oxygen atoms (see Figure 5). It was also proposed that the MESP value at the minimum indicates the strength of the lone pair. The MESP topology of lone pair-bearing molecules viz. H_2_O, CH_3_NH^∙^ and NO_3_^−^ is illustrated in Figure 6. Some molecules showing electron localization not associated with lone pairs (viz. benzene, methane and cyclopropane) are also depicted. The CPs for these molecules do not follow the above criteria for lone pairs. Bijina et al. [91] observed that one of the three eigenvalues of the Hessian matrix at MESP minima is more positive (λ_max_) than the other two eigenvalues. λ_max_ also shows a strong linear correlation with the MESP value at the minimum, viz. *V*_min_ indicating the highly directional and localized nature of the lone pairs. The nature of electronic localization in experimental organic electrides has been explored by Kumar and Gadre [92] to understand the characteristics of trapped electrons as well as the reason for their low thermal stability. A single molecular unit extracted from the crystal structure revealed intriguing features of electrides correlating with their stability. These include the unusually deep MESP minimum located far away from its van der Waals (vdW (van der Waals)) surface and the isotropic behavior of trapped electrons revealed by the comparable magnitudes of eigenvalues of the respective Hessian matrix.

In a very recent work, Kumar and Gadre [93] explored the gradient vector field of MESP, ∇*V*(**r**), which has remained relatively unexplored in quantum chemistry. Previously, the gradient vector field of the MESP was used by Popelier [94] to calculate the atomic charges via the divergence theorem. The gradient of MESP, equivalent to the internal electric field of the molecule, is expected to unravel the dynamic aspects of chemical reactivity. Espinosa et al. [95], in an early work, used the electrostatic potential gradient determined from high-resolution single-crystal X-ray diffraction data to reveal the position of zero-flux surfaces and critical points. Electrophilic and nucleophilic sites are interpreted as influence zones delimited by zero-flux surfaces containing saddle points which reveal the path for preferred attack on reactive sites with finite influence zones. In MED-based AIM, Bader had utilized the zero flux surfaces (ZFS) of the gradient vector field of electron density, ∇ρ(**r**), for partitioning of molecular space into atomic basins. In a similar spirit, Kumar and Gadre [75] attempted a three-dimensional MESP-based partitioning of molecular space on the basis of the zero-flux surface of ∇*V*(**r**) to propose MESP-based AIM. The ZFS of ∇*V*(**r**) is found to enclose electron-rich atoms or groups of atoms in the molecule as against its MED counterpart. Figure 7 presents the MESP-based ZFS surfaces of N_2_, CO, OF^∙^, H_2_O, NH_3_BH_3_, singlet carbene (:CH_2_) and H_2_CO. The ZFS of homonuclear diatomic molecules such as N_2_ symmetrically bisects the bond, with none of the atoms having a closed basin, while heteronuclear diatomic molecules such as CO display a closed ZFS around the more electronegative atom. Anions possess an overall ZFS that enclose the whole system, with the net charge residing outside the surface. Intriguingly, in some neutral systems such as singlet carbene, none of the atoms possess a completely closed ZFS. It has also been noticed that the nature of the atomic basins brings out the asymmetric electronic distribution of the molecule. The closed nature of ZFS of ∇*V*(**r**) correlates with the shielded nature of the atom and low NMR chemical shifts as well as enhanced susceptibility for electrophilic attack, whereas an open ZFS represents deshielding of the corresponding atom/s [96].

## 4. Applications of MESP Topology to Face Selectivity, Cation Binding and Reaction Mechanisms

The role of electrostatic effects in determining the facial selectivities of π-systems (π-facial selectivity) in nucleophilic and electrophilic additions to trigonal carbon centers has been explored using the topological analysis of MESP. In a combined experimental and theoretical investigation on electrophilic in addition to 7-isopropylidenenorborane and 7-methylenenorborane derivatives, Mehta et al. [97,98] used MESP analysis to show that the presence of electron-withdrawing groups such as -CN and -COOMe in the *endo* position can promote syn-facial electrophilic addition. They also used MESP topography analysis to study the face selectivity in electrophilic additions to the 2-methylene-bicyclo [2.1.1] hexane system and showed that the selectivity is due to the interplay of electrostatics and Cieplak-type orbital effects [99]. In yet another study by Mehta et al., the relative importance of through-space, through-bond and electrostatic interactions was discussed to interpret the face selectivity in electrophilic additions to methylenenorsnoutanes [100].

Gadre and co-workers also utilized MESP topology to investigate the electrostatic principles underlying molecular reactivity. Suresh et al. [101] used MESP topography to understand the structural properties and subtle electronic effects in various chromium-substituted arene carbonyl complexes. It has been shown that the substituent effects are strongly felt on the MESP surrounding the carbonyl oxygen rather than the arene ring, which is highly deactivated due to its complexation with a strongly electron-withdrawing Cr(CO)_3_ moiety. MESP topography was also used for understanding the regiospecificity of Markovnikov reactions [102]. Suresh et al. introduced a classification of substituents into electron donating and withdrawing in various substituted ethylenes on the basis of the increase or decrease in the negative character of V_min_ compared to unsubstituted ethylene. With electron-donating substituents, V_min_ is located close to the unsubstituted carbon of the ethylenic unit, facilitating the π-complex formation of HCl with the carbon. Such a regiospecific π-complex favors the formation of Markovnikov-type transition state for the addition of HCl to CH_2_CHR. In the case of electron-withdrawing substituents, the V_min_ is less negative and located farther away from the ethylenic carbon atoms leading to the formation of less regiospecific π-complexes. The acidity of Brønsted acids has been interpreted and quantified based on the MESP of the corresponding conjugate bases/anions [103].

Balanarayan et al. [104] used MESP to elucidate reaction mechanisms and observed a direct correspondence between MESP topographical analysis along the reaction path for dipolar cycloaddition (DC) reactions (Figure 8). The efficacy of MESP topography in elucidating the electronic mechanisms of concerted reactions, as revealed by the curved arrows by an organic chemist, was clearly revealed via the analysis of two 1,3-DCs, namely, HCNO + HCCH and NNCH_2_ + HCCH. In the particular case of the 1,3-DCs wherein the literature finds multiple mechanisms from wave function-based methods, the MESP topography-derived mechanism gave a single consistent one. Such an MESP-based analysis, being an MED-based one, also circumvents the subjectivity involved in a wave function-based analysis.

The structure, reactivity and aromaticity of linearly annelated acenes and their BN analogues were investigated using MESP analysis [105]. The MESP topography patterns of acenes differ substantially from their BN-analogues, with BN-acenes showing more localized π electron features compared to acene analogues. MESP topography also revealed an overall lowering of aromaticity in the annelated systems explaining the ‘aromatic dilution effect’.

Suresh and co-workers widely utilized MESP as a tool for predicting the stabilities and reactivities of a variety of organometallic catalysts including pincer catalysts [106], first-generation Grubbs olefin metathesis catalysts [107] and various water splitting catalysts [108]. Recently, Anjali and Suresh [109] investigated the oxidative addition of halobenzenes and toluene to monoligated palladium catalysts and proposed an MESP-based electronic parameter to predict the ability of palladium complex to undergo oxidative addition.

MESP topology has also been proposed as a predictive tool for studying cation coordination to anionic systems. Electrostatics is known to play a dominant role in these interactions, while polarization effects may also play some role in energetics. The negative-valued MESP CPs of anions act as cation attractors and the MESP value at the CP indicates the strength of such interactions. Using a topography based electrostatic docking model, Gejji et al. [110] investigated ion pair formation in trifluoromethanesulfonate ion (Tf¯) with cations viz. Li^+^, Na^+^ and NH_4_^+^ and showed that MESP can predict different minima, transition-state and saddle point structures of the ion pairs on the PES. MESP topography has also been employed for the investigating the structure and properties of various ion pairs such as M^+^BF_4_¯(M = Li or NH_4_), (CF_3_SO_2_)_2_N¯Li^+^ and Li^+^-(diglyme) [111,112,113].

## 5. Electrostatic Potential for Intermolecular Complexation (EPIC) Model and Its Applications

By exploring the MESP topology of weakly interacting molecules, Gadre and co-workers established that the negative-valued CPs in the MESP topology of a molecular species can act as anchoring points for the positively charged atoms in another interacting molecule. This results in a complementary electrostatic “lock-and-key” type of binding of the two species. The strength of the interaction depends on the extent of charge concentration at the CPs reflected by their corresponding MESP values. This observation indicates that there exists a strong relationship between molecular recognition and the corresponding three-dimensional MESP distribution. In 1997, Gadre and Pundlik proposed a new model, viz. electrostatic potential for intermolecular complexation (EPIC) [114,115], predicting the interaction patterns between any two species using the MESP features of them. The first step in this model involves the location and characterization of the CPs of the two monomers followed by calculation of MESP-driven nuclei-centered point charges using the individual molecular wave functions. The knowledge of MESP topography is then utilized for positioning the monomers A and B such that the hydrogen atoms in A is close to the MESP minima in B or vice versa. The interpenetration of the two species is avoided by embedding the heavy atoms in their respective vdW spheres. The EPIC interaction energy is estimated as
(13)$$E=1/2\left\{\sum V_{A,i}q_{B,i}+\sum V_{B,i}q_{A,i}\right\}$$
where V_A,i_ is the MESP value due to A at the *i*^th^ atom of the molecule B and q_B,i_ is the MESP-derived charge at this site. The interaction energy is minimized by keeping either A or B fixed and rotating and translating the other molecule in all possible directions without allowing the monomers to cross each other’s vdW boundaries. EPIC model thus generates a binary complex utilizing the spatial characteristics of MESP of the two monomers.

Gadre and Pundlik demonstrated the appropriateness of the model for exploring DNA base pair and base trimer interactions involving adenine (A), guanine (G) and cytosine (C) [114,115]. The predicted interaction energies as well as geometrical patterns have been found to agree remarkably with the corresponding ab initio data obtained from full geometry optimization. Further, the EPIC model was useful for revealing the stepwise hydration patterns of the 18-crown-C6 (18C6) molecule [116]. This MESP-guided approach correctly predicted the hydrogen bond formation patterns of the incoming water molecule with the complementary atoms of the crown ether. The cooperative electrostatics-guided hydration process revealed from this work stimulated further attempts to interrogate stepwise hydration patterns, structures and energetics of other interesting molecular species including uracil and formamide [117,118]. In all the cases, the most negative-valued MESP CPs served as a harbingers to the interacting water molecules. In summary, the EPIC model provided a rational approach for understanding the hydration patterns as well as the hydration shell structure. For instance, the hydrated structures of uracil viz. U…*n* (H_2_O) were found to incorporate square and cubic patterns of water molecules.

The EPIC model was also employed for predicting the structures and energetics of a large variety of weakly bonded complexes at negligible computational effort. Jovan Jose and Gadre [119] investigated the geometric patterns of (CO_2_)*_n_* clusters, with *n* = 2–8, using the EPIC model for generating energetically favorable initial geometries for subsequent MP2 and DFT-based calculations. The clusters with more C=O…C interactions were found to be associated with higher stabilization energies. This work was further extended to large (CO_2_)*_n_* and CO_2_Ar_m_ clusters in conjunction with EPIC [120]. The MESP-guided approach to the study of cluster formation was successfully applied to ZnS, C_6_H_6_, NO_2_, H_2_O, and NaCl clusters, thus opening up the possibility of investigating large-sized molecular clusters with high level ab initio methods using minimal computer hardware [121]. 

## 6. MESP Topography, π Systems and Substituent Effects

Gadre and Suresh utilized MESP topography to explain a variety of chemical phenomena including aromaticity, resonance effects, inductive effects and proximity effects. The substituent constant (σ), introduced by Hammett [122], is a measure of the electronic perturbations introduced by the substituent on the reaction center. The dual parametric equation, viz. $\sigma_{P}=\sigma_{I}+\sigma_{R}$, proposed by Taft [123], breaks up the total substituent effect ($\sigma_{p}$) as a combination of inductive ($\sigma_{I}$) and resonance ($\sigma_{R}$) effects. Politzer and co-workers extensively used MESP for understanding general electrophilic substitution reactions, particularly those involving substituted benzenes and derived linear correlation between most negative MESP and Hammett constants of the substituents [25,37,124]. Gadre et al. had showed that, in substituted benzenes, MESP minima are seen over the *ortho* and *para* positions for electron-donating substituents [125,126,127]. On the other hand, MESP minima are observed over the *meta* position in the case of electron-withdrawing substituents. The MESP minima observed over the meta and para positions showed excellent linear correlations with the σ meta and para constant, respectively.

Figure 9 illustrates the effect of substituent over the π-region in various carbon atoms of a phenyl ring. There is a reduction in the distance of the CP from the ring plane as well as a lowering in the corresponding MESP minima values compared to benzene in the case of electron-donating substituents. A reverse trend is seen for electron-withdrawing substituents. In a brief study, Gadre and Suresh [125,126,127] demonstrated the utility of the MESP topographical approach for the quantification of substituent effects for singly substituted benzene. In a further detailed investigation, they observed good linear correlations between *V*_min_ and experimentally observed Hammett σ values of *ortho*- and *meta*-disubstituted benzenes [126]. The MESP CP data also clearly reflected the effect of different orientations and conformations as well as the activation–deactivation effect of the substituents on the benzene ring. For quantifying the simultaneous effect of two substituents on benzene, Suresh and Gadre introduced substituent-pair constants viz. *D*_p_, *D*_m_ and *D*_o_, respectively, for *para*, *meta* and *ortho* arrangements with the benzene ring. The *V*_min_ values of triply substituted benzenes, predicted using the substituent-pair constants, showed a remarkable agreement with the values obtained at the HF/6-31G (d, p) level.

Later, Galabov et al. [128] proposed electrostatic potential value at the carbon atoms, *V*_c_, in the *para* and *meta* positions of substituted benzenes as a descriptor of substituent effect. From a study of substituted benzenes, Suresh and Gadre [127] confirmed that both *V*_c_ and *V*_min_ are good descriptors of substituent effect, showing a good linear relationship with the substituent constant σ^o^. However, compared to the nuclei-centered quantity *V*_c_, *V*_min_ clearly reflects the location of electron-rich regions of molecules in three dimensions. The energetics of cation–π interaction obtained from an isodesmic reaction scheme of the benzene derivatives with Li^+^ also showed a good linear correlation with the calculated *V*_min_ values, further confirming the reliability of *V*_min_ as a measure of the substituent effect.

MESP has also been used as an effective descriptor for the quantitative assessment of inductive and resonance effects. Suresh et al. [129] obtained a linear relation between *V*_min_ and the inductive substituent constant $\sigma_{I}$ by investigating the MESP properties of various 4-substituted bicyclo [2.2.2] octane carboxylic acids and quinuclidines using DFT calculations. The additivity effect of $\sigma_{I}$ has also been validated with the *V*_min_ approach using these multiply substituted derivatives. A *V*_min_-based methodology [130] for separation and quantification of through space (TS) and through bond (TB) effects has also been proposed. From a number of studies, Sayyed and Suresh established methods based on MESP to quantify proximity effects, the additive nature of substituent effects in multiply substituted systems [131] and substituent effects on cation–π interactions [132,133]. In a recent study, Remya and Suresh [134] showed that the difference between MESP at the nucleus of the para carbon of substituted benzene and a carbon atom in benzene is useful for quantifying substituent effects.

A rigorous and general definition of aromaticity, a concept closely associated with the extent of electron delocalization in a molecule, has remained elusive in the literature despite numerous experimental and theoretical attempts. Clar’s aromatic sextet theory [135] has been instrumental in defining the aromaticity of polycyclic benzenoid hydrocarbons (PBH) on the basis of the maximum number of sextets viz. six π-electrons represented by a circle. Motivated by their previous works on substituted benzenes, Suresh and Gadre [136] revisited Clar’s sextet theory using MESP and showed that an elaborate characterization of the π-regions of PBHs is possible from the number, nature and distribution of CPs in the MESP topography. Benzene possesses the signature of a perfect π-delocalization, with six identical (3, +3) CPs on either side of the ring with two adjacent minima connected via a saddle point. The average value of MESP at the CPs observed on one side of the ring was proposed as the measure of aromaticity. In comparison with benzene, each ring of the annelated systems showed three or less (3, +3) CPs with varying degrees of smaller π-delocalization. The MESP topology-based local and global aromaticity values correlated well with the earlier theoretical results by Li and Jiang [137] and Zhou and Parr [138]. Thus, the aromatic characterization of each ring in a PBH system is made feasible by simply looking at the MESP CP distribution. Vijayalakshmi and Suresh [139] analyzed a series of PBHs and demonstrated that the MESP values at the ring critical point (*V*_rcp_) of the π-electron cloud of each ring of PBH provides a good local measure of aromaticity. Further, MESP maps were found to provide useful pictorial representations of Clar’s theory.

Very recently, Anjalikrishna et al. [140] proposed a MESP CP-based approach to quantify the localized and delocalized π-electron distribution in aromatic, antiaromatic, nonaromatic, annulene and hybrid systems. In general, MESP CPs lie interior to the six-membered rings in PBHs (Figure 10), while the CPs are located exactly on the top and bottom of the π-regions in linear polyenes and annulenes. The MESP CPs are embedded in gradient paths and they lie outside the boundary of the rings in antiaromatic systems and hybrid systems consisting of aromatic and antiaromatic moieties [140]. The aromatic character of cyclic structures is defined in terms of the eigenvalues of the Hessian matrix viz. λ_1_, λ_2_ and λ_3_ at MESP minima and the imperfection in the aromatic character of a PBH system is measured as the deviations with respect to the corresponding eigenvalues of the benzene molecule. The eigenvalues were noticed to follow the trend λ_1_ ≫ λ_2_ > λ_3_ ≅ 0 in PBHs, λ_1_ > λ_2_ > λ_3_ ≅ 0 in linear polyenes, and λ_1_ > λ_2_ > λ_3_ ≠ 0 in antiaromatic systems. MESP topology-based analysis by Gadre, Suresh and their co-workers has thus provided a new dimension to the concept of aromaticity along with new ways to compare local and global aromaticity values of different molecules. The MESP topology analysis provided an in-depth assessment of localized or delocalized distribution of π-electrons in cyclic, acyclic, and strained-cyclic unsaturated hydrocarbon systems.

## 7. MESP as a Tool for the Quantification and Characterization of Noncovalent Complexes

MESP is a powerful tool in understanding and rationalizing non-covalent interactions. Kollman et al. [141] proved the existence of a good correlation between H-bond energies and the magnitude of MESP at a fixed distance from the proton acceptors in a series of dimers formed between HF and various acceptors. Espinosa et al. [142] analyzed the electrostatic potential in the intermolecular and hydrogen bonded regions in asymmetric units of l-arginine phosphate monohydrate (LAP) calculated from experimental X − (X + N) and theoretical ab initio SCF models. They showed a qualitative agreement between the models of V(**r**) obtained from experimental and theoretical calculations within 0.05 e Å^−1^ in the region of N1−H_4_…O_2_ hydrogen bonds between the arginine and phosphate groups of LAP. Later, Mata et al. [143] showed that the electrostatic potential in hydrogen bonding regions shows topological features complementary to electron density distribution. They observed zero-flux surfaces of electrostatic potential containing a saddle point (analogous to the bond critical point of electron density distribution) which separates the hydrogen bonding region into two parts viz. two enclosed regions or basins where the integrated net charge vanishes. It has also been shown that the topological properties of both ρ(**r**) and V(**r**) show similar dependences with the N–H distance at bond critical points.

Galabov et al. [128,144,145,146] demonstrated the use of MESP at atomic sites as a reactivity descriptor for the study of hydrogen bonding. In an early work, Gadre and Bhadane [147] explored MESP topography for understanding the primitive binding patterns in hydrogen-bonded dimers of HF with other small molecules (MY) at the Hartree–Fock level. They noted a linear relationship between the hydrogen bond length (r_HYD_) for the MY–HF complex and the distance between atom Y and most negative MESP point (r_ESP_). They proposed that r_HYD_ being basically the vdW contact distance Y–H can be used for the estimation of vdW radii of Y and H atoms. They also investigated [148] the hydrogen-bonded complexes of monosubstituted aliphatic carbonyl compounds (HCOR: R = F, Cl, CN, OH, SH, NH_2,_ CH_3_, CF_3_, NO_2_) with hydrogen fluoride to show that the strength of MESP at the CPs of carbonyl molecules serve as a good indicator of the strength of the respective hydrogen bonding interactions. Politzer and co-workers [149,150,151,152] proposed and popularized the use of molecular surface electrostatic potential (MESP computed on 0.001 a.u.-valued MED surface) and its positive and negative extrema to analyze intermolecular interactions, particularly hydrogen bonds and halogen bonds. Roy et al. [153] used MESP topography to interpret how molecules recognize each other at large intermolecular separations. Sayyed and Suresh [132,133] used MESP critical features to describe cation–π interactions. Additionally, MESP at nuclei [154] are used to derive the strength of a variety of hydrogen-, halogen-, and dihydrogen-bonded complexes.

Gadre et al., utilized MESP topographical information to probe lone pairs and interpret the chemical reactivity of molecular systems. Lone pairs play a ubiquitous role in determining molecular structure, reactivity and interactions. Electron-deficient aromatic rings with halo-, cyano- and nitro-substituents have an inherent tendency to interact with lone pairs of neutral molecules and anions, giving rise to lone pair–π or anion–π interactions. Mohan et al. [155] used MESP features to characterize and quantify lone pair–π interactions, employing the complexes formed between the lone pair regions of a variety of neutral, anionic and free radical systems with electron-deficient π systems including hexafluorobenzene (HFB) (Figure 11), 1,3,5-trinitrobenzene, 2,4,6-trifluoro-1,3,5-triazine and 1,2,4,5-tetracyanobenzene. Optimized geometries of a set of HFB–lone pair complexes are depicted in Figure 11. The location and the value of MESP *V*_min_ enable [116] a prediction of the orientation of the lone-pair-bearing molecule when it complexes with an electron-deficient system as well as the strength and directionality of the lone pair–π interactions. The excellent linear correlation between the interaction energies of the lone pair–π complexes and *V*_min_ values of the electron-rich systems suggests the ability of *V*_min_ to provide a priori prediction of lone pair–π interaction energy. Use of the EPIC model confirmed the dominance of electrostatic effects in controlling the orientation of the lone-pair-bearing molecules on the electron-deficient host systems as well as the interaction energy of the lone pair–π systems. This work also brought out the remarkable ability of electron-deficient aromatic rings to effectively ‘sense’ the lone-pair-bearing species. Further, Mohan and Suresh [156] designed receptors using HFB scaffolds for binding neutral molecules and anions predominantly through lone pair–π interactions. Bijina et al. [91] extended this work to investigate the topographical features of lone-pair-bearing organic molecules and inorganic ligands and their interactive behavior with HF, CO_2_ and Li^+^. The interaction energies correlated well with the respective *V*_min_ values in all the complexes of HF, while some minor deviations occurred in the complexes of CO_2_ and Li^+^ owing to other competing secondary interactions as well as the influence of multiple lone pairs on the same atom. They also noted that the directionality of lone pairs quantified in terms of distance and angle parameters dictates the directionality of the corresponding lone pair–HF interactions.

## 8. Concluding Remarks and Future Outlook

Molecular electron density (MED) and its sister scalar field, viz. the molecular electrostatic potential (MESP), are connected by a simple relationship. For a given molecular framework, the MESP is calculated by subtracting out the electronic contribution from the corresponding positive nuclear potential (Equation (5)). Similarly, MED at a reference point is proportional to the Laplacian of the MESP at that point, through the Poisson equation (Equation (8)). A major difference between these two scalar fields is that the MED assumes only non-negative values at all points in space, whereas the MESP at a point may attain a positive, zero or negative value. The topological features of MED were extensively explored by Bader and co-workers to understand the structure, bonding and reactivity of molecules. However, in spite of the direct relationship between these two scalar fields mentioned above, the topological studies of MESP for unearthing the bonding and reactivity patterns of molecules were evident by their absence until 1990.

The scalar field MESP is endowed with rich topological features, such as minima and saddle points. MESP attains positive-valued critical points (CPs) in the vicinity of the nuclear framework. However, the most interesting topological signatures of MESP lie in the outer regions, and exhibit negative-valued CPs. The algorithms developed by Gadre and co-workers for rapid topology mapping of MESP and its CPs [71,83,84] catalyzed the detailed studies of electrostatic features of molecules. These features have been investigated during the last three decades by the authors’ research groups for shedding light on a variety of chemical concepts and phenomena. The present review article has summarized such works on the sites of electrophilic attack [98,110], cation binding sites [110,111,113], anionic sizes and shapes [56,157], lone pairs [90,91], π bonds [102], aromaticity [136,140], substituent effects [125,126], lone-pair–π interactions [155,156], etc.

An advantage that MESP has in elucidating reaction mechanisms and molecular recognition is the distinct negative character of the function at electron-rich regions which directly connects with chemical concepts such as π bonds, lone pairs, etc. Thus, the use of MESP topology along a reaction path offers a direct correspondence between the electronic shifts and the curved arrows representation suggested by Robinson [158] drawn in depicting electronic mechanisms. Particularly noteworthy ones are MESP-based signatures of molecular recognitions [153] and elucidation of organic reaction mechanisms [102,104]. We have only carried out a few exploratory studies in these areas, which have been quite successful. It would be worthwhile to take up more exhaustive studies in these two areas. The advantage of MESP-based investigations is that they are based on the molecular electron density distribution and do not use the molecular orbitals, which are not uniquely defined.

MESP is known to be a valuable tool for predicting directionality of approach of two interacting molecules. The present article has summarized the electrostatic potential for intermolecular complexation (EPIC) model developed by Gadre and co-workers [114,115]. This model has been generalized and used extensively for assembling molecular aggregates [116,120]. It has been recently generalized into a molecular cluster building algorithm [121]. This approach would be immensely beneficial for building up large molecular clusters and would shed light on the atomic contact patterns in molecular crystals. We anticipate that it may serve as a predictive tool for the major challenge of crystal structure prediction.

Another promising area of future research is MESP-based AIM, recently proposed by Kumar et al. [93,96]. It brings out the anisotropy in the gradient paths for the molecule under consideration which could be employed for understanding the stereodynamics of the interacting molecules. The MESP-based zero flux surfaces vividly bring out the electron-rich regions of the molecules. The shielding effects affecting the chemical shifts in NMR are also clearly displayed by such analysis. The analysis of gradient paths goes beyond the examination of only the scalar features of MESP and holds enormous potential for understanding the molecular properties and the reactive patterns. It may be pointed out that there are some other approaches for studying the reactivity of molecules. These do not explore the topological features of the molecular electron density and electrostatic potential. Many of them are based on the so-called conceptual density functional theory [159,160,161,162,163,164,165,166].

In this article, we have attempted to summarize the prowess of the topological analysis of the molecular electrostatic potential for exploring a variety of physicochemical phenomena. It is astounding that a single scalar field can be harnessed for unravelling such diverse features of the molecular world.

## Figures and Tables

**Figure 1 molecules-26-03289-f001:**
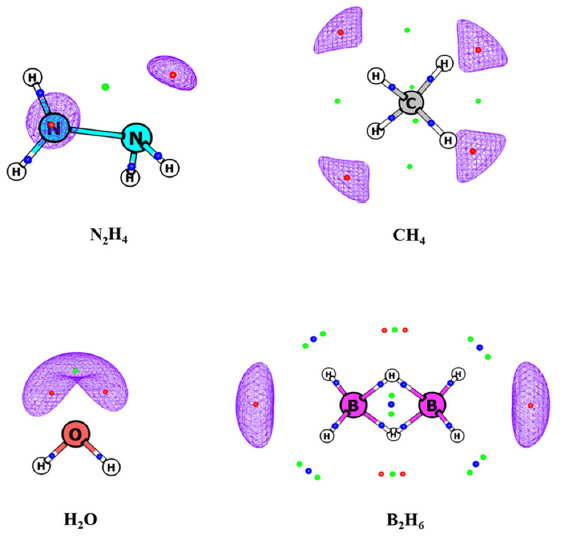
MESP topology of some molecules obtained from B3LYP/6-311G (d, p) calculations. Blue, green and red dots correspond to (3, −1), (3, +1) and (3, +3) critical points (CPs). Blue dots on the bonds are (3, −1) bond CPs. Isosurface values are −60, −2, −50 and −4 kcal/mol for N_2_H_4_, CH_4_, H_2_O, and B_2_H_6_, respectively.

**Figure 2 molecules-26-03289-f002:**
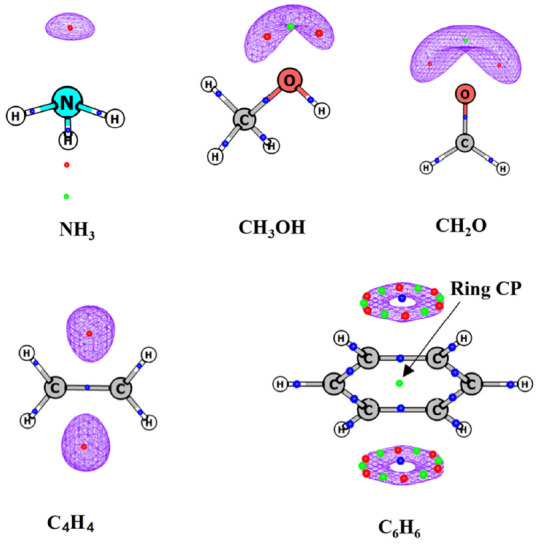
MESP topology of some molecules obtained from B3LYP/6-311G(d, p) calculations. Blue, green and red dots correspond to (3, −1), (3, +1) and (3, +3) critical points (CPs). Isosurface values are −60, −40, −30, −15, and −10 kcal/mol for NH_3_, CH_3_OH, CH_2_O, C_2_H_4_, and C_6_H_6_, respectively.

**Figure 3 molecules-26-03289-f003:**
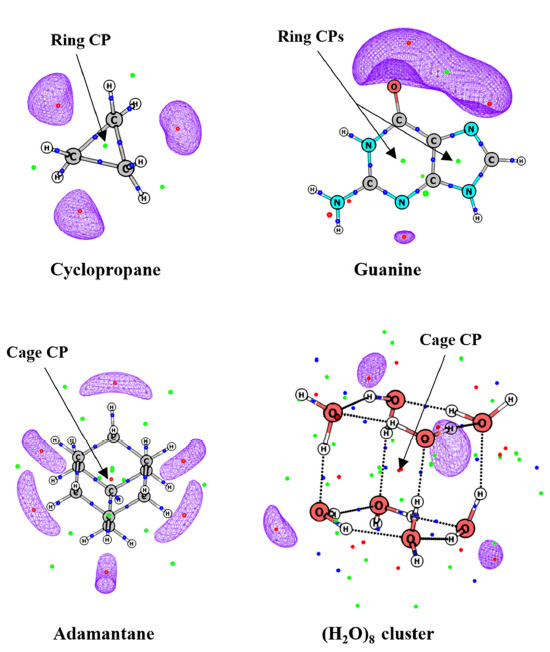
MESP topology of some molecules obtained from B3LYP/6-311G (d, p) calculations. Blue, green and red dots correspond to (3, −1), (3, +1) and (3, +3) critical points (CPs). Isosurface values are −10, −40, −2, and −10 kcal/mol for cyclopropane, guanine, adamantine and (H_2_O)_8_ cluster, respectively.

**Figure 4 molecules-26-03289-f004:**
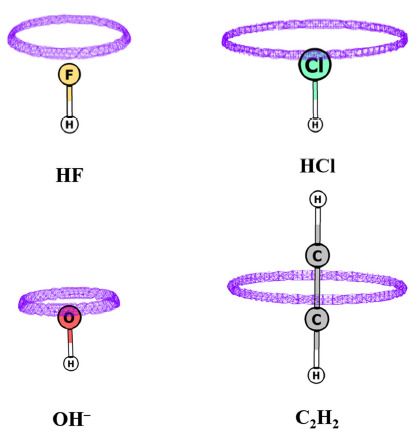
MESP topology of some molecules calculated at B3LYP/6-311G (d, p) level theory. Degenerate ring of CPs of (2, 0) type lies inside the isosurface for each molecule. Isosurface values are: −30, −10, −280, −20 kcal/mol for HF, HCl, HO¯ and C_2_H_2_, respectively.

**Figure 5 molecules-26-03289-f005:**
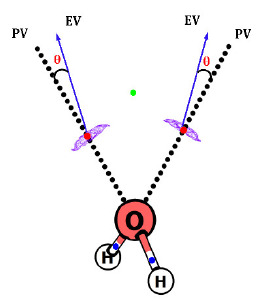
Direction of the eigenvector, EV, corresponding to the largest eigenvalue (blue arrow) and position vector, PV (dotted line), for H_2_O at MP2/6-311++G(d, p) level theory. θ is the angle between EV and PV is denoted as θ. See text for details.

**Figure 6 molecules-26-03289-f006:**
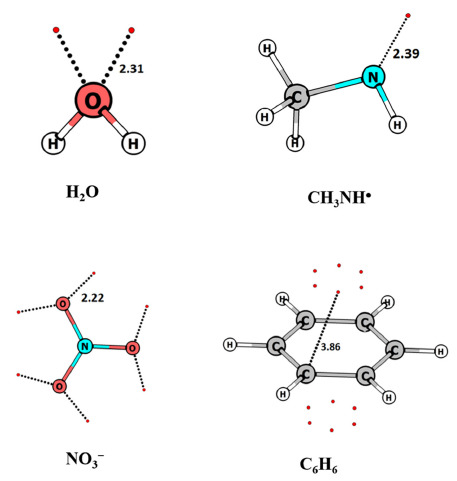
MESP topography of H2O, CH3NH^∙^, NO_3_**^−^** and C_6_H_6_ employing MP2/6-311++G(d, p) calculations. Minima shown by red dots. The distance is in Å.

**Figure 7 molecules-26-03289-f007:**
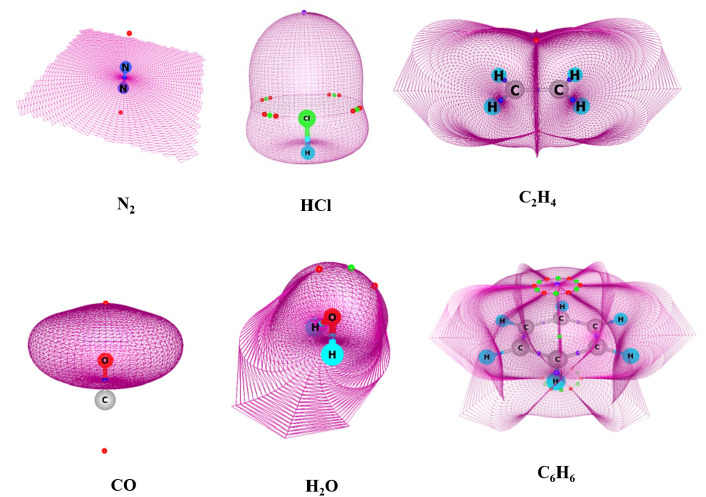
MESP-based ZFS surfaces of N_2_, HCl, CO, C_2_H_4_, H_2_O and C_6_H_6_ NH_3_BH_3_ employing B3LYP/6-311G (d, p) calculations. Color-coded dots for CPs are: red (3, +3), green (3, +1) and blue (3, −1). See text for details.

**Figure 8 molecules-26-03289-f008:**
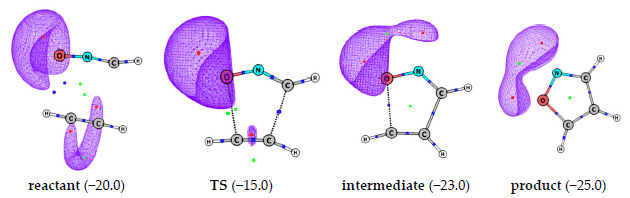
MESP topography and isosurfaces along the reaction path for the 1,3-dipolar cycloaddition between HCNO and HCCH, at B3LYP/6-311++G (d, p) level theory. Red, blue and green dots correspond to (3, +3), (3, −1) and (3, +1) CPs, respectively. Isosurface values in kcal/mol, in parentheses.

**Figure 9 molecules-26-03289-f009:**
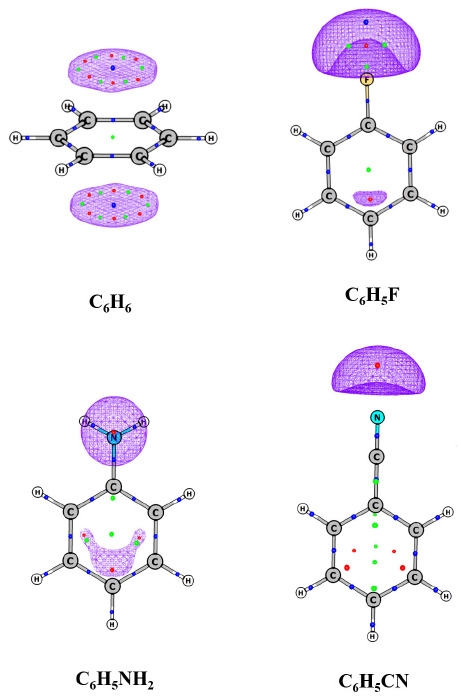
MESP isosurfaces of benzene and monosubstituted benzenes. The isosurface values are −10, −10, −20, −30 kcal/mol for C_6_H_6_, C_6_H_5_F, C_6_H_5_NH_2_ and C_6_H_5_CN, respectively. B3LYP/6-311G (d, p) calculations. See text for details.

**Figure 10 molecules-26-03289-f010:**
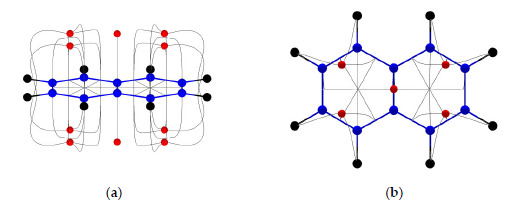
MESP topography of naphthalene molecule, calculated using B3LYP/6-311+G(d, p) level theory. (**a**) Side view and (**b**) top view of (3, +3) CPs (shown by red dots) embedded in gradient paths shown by grey curves. Red dots are MESP minima. See text for details.

**Figure 11 molecules-26-03289-f011:**
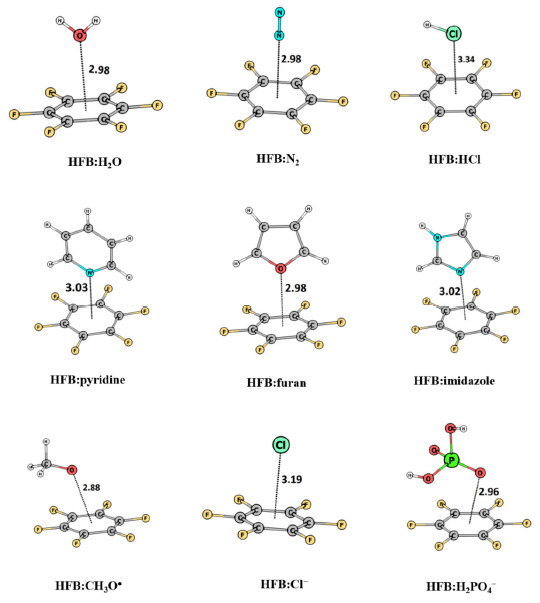
Hexafluorobenzene (HFB)-lone pair complexes along with the indicated interaction distances in Å at M06L/6-311++G (d, p) calculations. See text for details.

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
