# Peer review of "Electrostatic Potential Topology for Probing Molecular Structure, Bonding and Reactivity"

_molecules, 2021, doi:10.3390/molecules26113289_

Round 1

Reviewer 1 Report

Electrostatic Potential Topology for Probing Molecular Structure, Bonding and Reactivity

By Gadre et al.

Over the last 3 decades Gadre and co-workers have indeed made a long standing and unique contribution to the title topic of this well-written review. It is nice to see such a (perhaps nowadays rare) sustained output discussed in this comprehensive and didactical way. Guided by nice pictures and a flowing text, the reader is shown a rich and interesting tour in the world of the topology (and indeed not just the topography, an old and unfortunately recurring misnomer) of the electrostatic potential. This group has thus added a strong and useful contribution to the ever growing area of Quantum Chemical Topology (QCT), a field that deserves to be named and cited in the manuscript to clarify the context of this important work, and thereby expose and promote it (see below for details). All in all, it is surprising that no work on Enrique Espinosa is mentioned in spite of the overlap of his work with this work. Some generosity is in place. Finally, the Introduction should be non-technical because novice readers will be put off by what may be perceived a non-gentle and even introspective introduction to material they may otherwise engage with. There should be new Section (“Background” or so) discussing all technical material, while the Introduction needs to be expanded and present the context of the work better (or at all actually).

Of course this review should be recommended for publication but the authors are kindly asked to act on the specific points below, which are preceded by the unique line number of the manuscript.

Improvements to be made:

  • 54: It is useful to show an example of degenerate CP: e.g. (2,0)  so that the reader gets used to this (r,s) notation.
  • 83: An AIL is not the same as a BP so the “or” is not right. In fact, the meaning of the BCP has been attacked much in the last few years. The proposed name Line Critical Point (LCP) is more neutral than BCP because it severs the tie between the critical point and its potential chemical meaning. In fact, the BCP is the only CP that has such a tie; in contrast, a RCP only describes the ring nature of the connection rather than refer to aromaticity, for example. Had it been called ACP (“aromatic critical point”), it could have ended up with the same troubled history as the BCP.
  • 118: the BNP has been linked to the electron density by investigating a homeomorphism between them: J.Quant.Chem., 109, 2542-2553 (2009).
  • 126: THE Pullmans
  • Somewhere in Section 1 (or new “2” actually) it is useful and proper to mention Comput.Chem., 24,437-442 (2003), which for the first time classified gradient paths by the type of critical points they connect. This knowledge is valid for any 3D scalar filed and could be applied to the MESP too. That paper also presents a general algorithm for the robust localisation of critical points.
  • Quantum Chemical Topology, a increasingly used name coined in 2003 by Popelier, is described in Box 8.1 of “The QTAIM Perspective of Chemical Bonding “, P.L.A. Popelier in “The Nature of the Chemical Bond Revisited”, Chapter 8, pp 271-308 (pp 1-38), Eds. G. Frenking and S. Shaik, Wiley-VCH, 2014. An excerpt from this box follows:  

“A non-exhaustive list of 3D topologically partitioned quantum mechanical functions includes:

  • Electron density ?(r) (started with Ref. [20]).
  • The Laplacian of ?, ∇2?(r) (started with Refs [68, 69] and full topology first

explored in Refs [70–72]).

  • (Bare) nuclear potential Vnuc(r) (early start with Ref. [73] but elaborate and

self-contained study [17]).

  • ELF [14] (started with Ref. [13] and reviewed in Ref. [74]).
  • Electrostatic potential [75] (started with thorough studies [76, 77] and continued

with Ref. [78], applied in the area of chemical reactions [79], Lewis acidity [80]

or electron diffraction study [81]).

  • Virial field (or trace of the Schroedinger stress tensor) (topology explored in Ref.

[57].).

  • Magnetically induced molecular current distributions (started with [82]).
  • Intracule density (started with Ref. [83], which reveals correlation cages).
  • Ehrenfest force field (topology first investigated [84] in 2012).
  • Energy partitioning (beyond the kinetic energy and atom virial theorem)

(Coulomb potential energy partitioning started with [37] and culminated into

the theory of IQA [24], leading to energetic underpinning for the topological

expression of chemical bonding [85]).”

The main author may be pleasantly surprised that refs.75 to 79 are all to his work.

  • 220: degenerate CPs are *not* unique to the MESP. They occur in the electron density and are discussed at length in Bader and Popelier’s book. They also occur in the topology of the Laplacian (of the electron density), even doubly degenerate (1,-1), e.g. Phys. Chem. A 2001, 105, 7638-7645.
  • 251: approach
  • 317: …localization: (i)….
  • 344 and thereabouts. It may be interesting to the authors that the gradient vector field of the MESP has been used to calculate atomic charges via the divergence theorem: Chem. Acts, 105, 393-399 (2001).
  • 388: Markovkinov
  • 572: use subscripts
  • 642: “its” cannot be right logically. If MESP is a sister, then it is female, and thus the MED is female because it is also a field, like the MESP. Thus “its” should be “her”.  
  • 666/667: how this negativity is key is a mystery. Explain or make more self-contained.

Author Response

File attached

Reviewer 2 Report

The review paper, although useful, lack of systematic reactivity analysis, being focused merely on topological electrostatically cause-effects of atoms in molecule(s). I hearty advice authors including such section or notes for which they may use the related references as MATCH Commun. Math. Comput. Chem 64 (2), 391-418; idem 66, 35-63; idem 60, 845-868. Having as such analysis inclusion performed the review may be reconsidered for its broaden interest in molecular phys-chem chem phys community!

Author Response

File attached

Reviewer 3 Report

This review summarizes the contributions of the authors' groups over the last three  decades and those of the other active groups towards understanding chemical bonding, molecular recognition, and reactivity through topology analysis of MESP.

I recommend to publish  the present version.

Author Response

File attached
